# Development and In Vitro Cytotoxicity Evaluation of Individual and Combined Injectable Solutions of Curcumin and Resveratrol Against Lung Cancer Cells

**DOI:** 10.3390/antiox14080983

**Published:** 2025-08-11

**Authors:** Ximena Hernández Martínez, Carla O. Contreras-Ochoa, Marisol Mir-Garcia, Nataly Aguilar-García, Hugo Cortés Martínez, Elvia A. Morales-Hipólito, Sandra L. Hernández-Ojeda, Mariana Dolores-Hernández, Bruno Solis-Cruz, J. J. Espinosa-Aguirre, Daniel Hernandez-Patlan, Raquel López-Arellano

**Affiliations:** 1Laboratory 5: Pharmaceutical Development Testing Laboratory (LEDEFAR), Multidisciplinary Research Unit, Superior Studies Faculty at Cuautitlan (FESC), National Autonomous University of Mexico (UNAM), Cuautitlan Izcalli 54714, Mexico State, Mexico; ximena.hernandezmt@comunidad.unam.mx (X.H.M.); eadriana_mh@comunidad.unam.mx (E.A.M.-H.); may34@comunidad.unam.mx (M.D.-H.); bruno_sc@comunidad.unam.mx (B.S.-C.); 2Infectious Disease Research Center, National Institute of Public Health, Av. Universidad No. 655, Santa María Ahuacatitlán, Cuernavaca 62100, Morelos, Mexico; ccontreras@insp.mx (C.O.C.-O.); marisol.mir@uaem.edu.mx (M.M.-G.); yessica.aguilarga@uaem.edu.mx (N.A.-G.); 3Waters Corporation, Benito Juárez, Mexico City 04510, Mexico; Hugo_Cortes@waters.com; 4Institute of Biomedical Research, National Autonomous University of Mexico (UNAM), Third Exterior Circuit, University City, Mexico City 04510, Mexico; slhernandez@iibiomedicas.unam.mx (S.L.H.-O.); jjea@biomedicas.unam.mx (J.J.E.-A.); 5Nanotechnology Engineering Division, Polytechnic University of the Valley of Mexico, Tultitlan 54910, Mexico State, Mexico

**Keywords:** curcumin, resveratrol, injectable solutions, lung cancer, A549 cells, BEAS cells, cytotoxicity

## Abstract

The objective of the present study was to develop injectable solutions of curcumin (CUR) and resveratrol (RES) for intravenous administration as a strategy to increase their solubility and stability, as well as to evaluate their cytotoxic potential, individually and in combination, on human lung non-small adenocarcinoma cells (A549 cells) and non-tumoral cells isolated from normal human bronchial epithelium (BEAS cells) to establish possible synergistic effects and potential therapeutic alternatives for lung cancer. Using factorial experimental designs, the components of the injectable CUR and RES solutions were selected, and their hemolytic potential was evaluated by a static method. In addition, combinations of injectable CUR:RES solutions (25:75, 50:50 and 75:25) were prepared from the individual ones, and their stability under refrigeration conditions and cytotoxic potential on A549 and BEAS cells were evaluated. The stability of the injectable solutions of CUR, RES and their different combinations was maintained for 3 months, except for the 25:75 combination of CUR:RES. Furthermore, the cytotoxic potential of CUR and RES on tumoral cells (A549) and non-tumoral (BEAS) cells was evaluated, indicating a dose-dependent effect; the combination of CUR:RES 50:50 and the combination of CUR:RES 75:25 presented synergistic effects in reducing cell viability. This study suggests that injectable solutions of CUR, RES and their combination for intravenous administration could be potential viable candidates and should be evaluated for their efficacy in animal models of lung cancer to establish new possible treatments.

## 1. Introduction

Lung cancer is classified into two main categories: small-cell lung cancer (SCLC) and non-small-cell lung cancer (NSCLC). However, SCLC is the most prevalent type of cancer, accounting for 80–90% of all cases, and its diagnosis is not promising, as people are usually diagnosed in late stages of the disease due to the lack of specificity of early symptoms, which leads to 50% of patients dying within a year after diagnosis and only just over 20% surviving to 5 years after diagnosis [1]. Although there are currently more monotherapy options for lung cancer, it is a fact that combined therapies have proven to be more effective. However, their effects can be variable, and there are many side effects arising from the therapy [1], not to mention that cancer cells can generate resistance by mechanisms that include (1) modification of the site of action of the drugs, (2) expression of drug pumps, (3) detoxification mechanisms, (4) decreased susceptibility to apoptosis, (5) greater capacity for repair of genetic material, and (6) altered proliferation [2]. In this context, natural products, specifically polyphenols, have been seen as viable alternative for use as chemotherapeutics, chemopreventives or adjuvants.

Curcumin (CUR) and resveratrol (RES) are two naturally occurring polyphenols obtained from the rhizome of Curcuma longa and from red grapes and some berries (mulberries, blueberries and peanuts), respectively [3]. These polyphenols are mainly known for their excellent antioxidant, anti-inflammatory, immunostimulant and anticancer properties [4]. In fact, in in vitro models it has been reported that CUR has a cytotoxic effect on A549 adenocarcinoma cells because it inhibits cell proliferation by damaging the DNA of the tumor cell, producing stress in the endoplasmic reticulum and activating mitochondrial apoptosis, as well as triggering the accumulation of reactive oxygen species (ROS) [5,6]. Furthermore, in BEAS cells (non-tumor cells), CUR has also presented a cytotoxic effect, but less so compared to A549 cells [7,8]. RES has also been reported to have greater cytotoxic effects in A549 cells than in BEAS cells [9,10]. This polyphenol inhibits cell viability because it stops the cell cycle in the G2 phase of mitosis, induces apoptosis by ROS and inhibits the Akt protein that regulates metabolic functions of cell growth and survival [11,12].

Although CUR and RES have shown good effects, CUR is characterized by its limited solubility in water (0.6 µg/mL), poor permeability and bioavailability, so it can be classified as a class IV molecule in the biopharmaceutical classification system (BSC) [13]. Furthermore, its stability is low, and as the pH increases, its degradation rate also increases [14]. In contrast, RES is a BCS class II molecule, meaning it has low solubility and high permeability [15]. Like CUR, RES is relatively stable under acidic conditions, and its degradation is exponential at pH above 6.8, as well as when exposed to light and when the temperature increases [16].

To overcome the limitations of these polyphenols and ensure their reproducibility in terms of effects, different strategies have been developed, such as their incorporation into micelles, micro/nanoemulsions, lipid and polymeric nanoparticles, liposomes, solid dispersions and non-covalent complexes/nanocomplexes [14]. In fact, the use of nanotechnology platforms is one of the first strategies considered to address the limitations of CUR and RES, and they are also promising because they can be targeted, providing specific treatments with fewer or no adverse effects [17]. However, among the most important limitations of nanotechnological systems are their reproducibility in terms of size and polydispersity and their scalability to an industrial level [18]. In this sense, the development of liquid formulations such as solutions can be viable alternatives because they are low-cost, since expensive pharmaceutical technology is not required to obtain them [19]. Furthermore, it has been reported that most BCS class IV drugs must be administered parenterally to achieve therapeutically effective concentrations [13].

Therefore, in the present study, injectable solutions of CUR and RES were developed as a strategy to increase their solubility and stability, as well as to evaluate their cytotoxic potential, individually and in combination, on A549 and BEAS cells to establish possible synergistic effects and potential therapeutic alternatives to be evaluated in animal models of lung cancer.

## 2. Materials and Methods

### 2.1. Chemicals and Reagents

Reference substances of CUR (65%, catalog number SLBM5931V) and RSV (≥99%, catalog number SLBV8562) were purchased from Sigma-Aldrich (St. Louis, MO, USA). Curcumin raw material (CUR, purity 55.47%) was purchased from Laboratorios Mixim S.A. de C.V (Naucalpan, Estado de México, Mexico), and resveratrol raw material (RSV, purity 98%) was obtained from Alfadelta S.A de C.V (Naucalpan, Estado de México, Mexico). Propylene glycol (PG), polyethylene glycol 400 (PEG 400) and lactic acid (88% purity) were obtained from Drogueria Cosmopolitan (Naucalpan, Estado de México, Mexico). Sodium lactate (60% purity) was obtained from Productos BR (Atizapan de Zaragoza, Estado de México, Mexico). Acetonitrile (HPLC grade) and absolute ethyl alcohol (reagent) were from JT Baker (Radnor, PA, USA). Finally, formic acid (88% purity) and sodium chloride (NaCl, 99% purity) were provided by Fermont (Monterrey, Mexico). RPMI (Roswell Park Memorial Institute 1640 Medium) was purchased from Gibco Life Technologies (Waltham, MA, USA). MTS solution (Cell Titre 96 AQueous One Solution) was purchased from Promega (Madison, WI, USA).

### 2.2. Solubility Studies of Curcumin (CUR) and Resveratrol (RES)

Solubility studies of both CUR and RES were performed in different cosolvents: propylene glycol, PEG 400 and ethanol. Briefly, 0.10 g of CUR (purity 55.47%) or RES (purity 98%) was added to 25 mL of each cosolvent. The suspensions formed were kept under magnetic stirring for 24 h at room temperature to reach equilibrium. Subsequently, the samples were centrifuged (Beckman Coulter Microfuge 20R, Brea, CA, USA) at 2292× *g* for 10 min at 4 °C, and the supernatants were filtered using a 0.20 µm nylon membrane filter (Nylon Acrodisc, Gelman Sciences Inc., Ann Arbor, MI, USA) for subsequent analysis by ultra-performance liquid chromatography (UPLC^®^).

### 2.3. Analytical Method for the Quantification of CUR and RES

Simultaneous quantification of CUR and RES was performed using an Acquity ultra-performance liquid chromatography (UPLC^®^) H-Class system equipped with a quaternary pump system, an autosampler injector and a PDA photodiode array detector (Waters, Milford, MA, USA). Chromatographic separation of RES and CUR was carried out using a BEH Shield^TM^ RP18 column (100 mm × 2.1 mm, 1.7 μm; Waters Corporation, Milford, MA, USA). The composition of the mobile phase consisted of a mixture of acetonitrile/1% formic acid (55:45) at a flow rate of 0.3 mL/min (isocratic flow). The sample injection volume was 5 μL with an analysis time of 7 min. Detection was performed at wavelengths of 425 nm and 305 nm for CUR and RES, respectively. Empower 3 software (Waters, 2010, Milford, MA, USA) was used for data acquisition and processing.

The analytical method was validated according to the International Conference on Harmonization (ICH) topic Q2 (R1) guideline “Validation of Analytical Procedure: Text and Methodology” [20], as well as following the Food and Drug Administration (FDA)’s guidance for industry, “Analytical Procedures and Methods Validation for Drugs and Biologics” [21]. The validation of the analytical method included the evaluation of the analytical parameters for the adequacy, linearity and precision of the system, specificity, influence of the filter, linearity, precision, accuracy and intermediate precision. The system presented linearity at concentrations between 2.5 and 25 µg/mL.

### 2.4. Development and Selection of Injectable CUR Formulation

The injectable CUR solutions were prepared by homogenizing the cosolvent mixture under magnetic stirring at 400 rpm (IKA mechanical stirrer, RW 20 digital with propeller stirrer R 1382, IKA-Werke GmbH & Co., Staufen, Germany), followed by the addition of CUR raw material until complete solubilization and enough water to bring the solution to the final volume.

Formulation selection was performed using a 2^k^ factorial experimental design, considering CUR (0.2% and 0.3%, *w*/*w*, considering the purity adjustment), PEG 400 (0% and 60%, *w*/*w*) and ethanol (10% and 30%, *w*/*w*) as factors and the CUR content in the formulation and pH as response variables.

#### Optimization of the Prototype Formulation

The prototype injectable CUR solution was prepared by first homogenizing a mixture of 60% PEG 400 and 30% absolute ethyl alcohol by mechanical stirring at 400 rpm (IKA mechanical stirrer, RW 20 digital with propeller stirrer R 1382, IKA-Werke GmbH & Co., Staufen, Germany). Subsequently, 1.7% CUR raw material (considering the purity adjustment) was added, and stirring was maintained until complete solubilization. Finally, a lactic acid–sodium lactate buffer solution was added to reach the final volume (8.3%) and a pH of around 5.6 in order to improve CUR stability. The resulting solution was immediately filtered through filter paper (Whatman No. 41, Whatman International, Ltd., Maidstone, UK).

### 2.5. Development and Selection of Injectable RES Formulation

Like the injectable CUR solution, the injectable RES solution was selected through a 3^k^ factorial experimental design. The factors considered in the design were PG (0%, 30% and 60%), PEG 400 (0%, 30% and 60%) and ethanol (10%, 20% and 30%). The amounts of RES (1.7%) and NaCl (0.8%) were kept constant in the design. The response variables considered to evaluate the design were RES content and pH.

#### Optimization of Prototype Formulation

The prototype injectable RES solution was prepared by homogenizing 60% PEG 400 and 20% absolute ethyl alcohol under mechanical stirring at 400 rpm. Subsequently, 1.7% RES raw material (considering the purity adjustment) and a NaCl solution were added to achieve a 0.8% concentration in the formulation. Finally, the resulting solution was brought to the final volume with Milli-Q water (Merck Millipore, Darmstadt, Germany) and filtered through Whatman 41 paper. The pH of this solution was 6.7.

### 2.6. Evaluation of the Hemolytic Potential of Injectable Formulations

The hemolytic potential of the prototype injectable solutions of CUR and RES was evaluated using a static method [22,23]. For this, human peripheral blood was obtained using EDTA vacuum tubes (BD Vacutainer, BD Diagnostics, Franklin Lakes, NJ, USA). Subsequently, the tubes containing whole blood were centrifuged at 2500 rpm for 10 min, and the plasma was pipetted off and discarded. The cell pellet (red blood cells) was washed three times with Sörensen’s buffer pH 7.4. Then, 10 mL of the cell suspension was transferred to a test tube, and 15 mL of Sörensen buffer pH 7.4 was added to reach a concentration of 40% *v*/*v*. To evaluate the hemolytic potential of the injectable solutions, 5 mL of the red blood cell suspension (40%, *v*/*v*) was placed in tubes, followed by 0.5 mL of the injectable formulation of CUR or RES, 0.5 mL of Extran^®^ MA 02 neutral concentrate (as a positive control, Merck, Darmstadt, Germany) or 0.5 mL of Sörensen buffer pH 7.4 (as a negative control). After incubating the tubes for 10 min at 37 °C, they were centrifuged again at 2500 rpm for 10 min, and 0.1 mL of the supernatant was placed in 5 mL flasks and brought to the final volume with Sörensen buffer pH 7.4. The samples were read spectrophotometrically at 541 nm in triplicate, and the percentage of hemolysis for each formulation was calculated with the following formula:$$Hemolysis\left(\%\right)=\frac{{Abs}_{S}-{Abs}_{NC}}{{Abs}_{PC}}\times 100$$
where Abs_s_ is the absorbance of the injectable CUR or RES solution, Abs_NC_ is the absorbance of the negative control (Sörensen buffer pH 7.4) and Abs_PC_ is the absorbance of the positive control (Extran^®^ MA 02).

Hemolysis values < 10% are considered non-hemolytic, while values > 25% hemolysis are considered hemolytic [22].

### 2.7. Evaluation of Injectable Formulations

In addition to the injectable solutions of CUR and RES, three formulations were prepared based on the individual injectable solutions using different percentages of CUR and RES, namely, 25:75, 50:50 and 75:25, to determine possible synergistic, antagonistic or additive effects. For this purpose, the necessary volumes of the injectable CUR and RES (100%) solutions were mixed to obtain the corresponding combination.

#### 2.7.1. Stability Studies Under Storage Conditions

Stability studies were carried out on injectable solutions of CUR, RES and their different combinations under the conditions established for the stability of drugs intended to be stored under refrigeration (5 ± 3 °C). Briefly, 0.5 mL of the injectable solutions from independent vials were appropriately diluted and analyzed at 0, 1 and 3 months in triplicate. The quantification of CUR and RES was performed by UPLC, as described in Section 2.3.

#### 2.7.2. Cytotoxic Effect of Injectable Formulations of CUR and RES

##### Cell Lines

A549 cells (human lung non-small adenocarcinoma cells, RRID: CVCL_0023, American Type Culture Collection, Rockville, MD, USA) and BEAS-2B cells (non-tumoral cells isolated from normal human bronchial epithelium, American Type Culture Collection, Rockville, MD, USA) were cultured in RPMI supplemented with 10% or 5% fetal bovine serum, respectively, and 1% Antibiotic–Antimycotic (100 U/mL penicillin, 100 µg/mL streptomycin and 0.25 µg/mL amphotericin B) at 37 °C under a 5% CO_2_ atmosphere. Culture medium was changed every third day.

##### Cell Viability Assay

A549 or BEAS-2G cells were seeded overnight in 96-well plates at a density of 15,000 or 14,650 cells/well, respectively. Subsequently, the cells were exposed to the injectable solutions of CUR, RES and different combinations of CUR:RES (25:75, 50:50 or 75:25) for 48 h; a negative control (without polyphenols) was included. Furthermore, cells were also exposed to the antineoplastic drug cisplatin at the IC50 dose (0.04 mg/mL and 0.025 mg/mL for A549 and BEAS, respectively) as a positive control. Cytotoxicity induced by the polyphenols and cisplatin was evaluated using 20 μL/well of MTS solution incubated with the cells for 2 h at 37 °C. Absorbance was measured by a microplate reader (ELx800, BioTek, Swindon, UK) at 490 and 690 nm. Cell viability is reported as the percentage of absorbance of treated cells relative to untreated control cells.

The synergistic, antagonistic or additive interactions between CUR and RES in the injectable solutions containing the combination of polyphenols were evaluated through the coefficient of drug interaction (CDI), which was calculated using the following formula:$$CDI=\frac{C}{A\times B}$$
where C represents the cell viability rate with the combination of CUR and RES in the injectable solutions; A and B represent the cell viability rates with CUR and RES, respectively. CDI values less than 1, equal to 1 and greater than 1 indicate synergistic, additive and antagonistic effects [24].

##### Representative Images of Cytotoxicity Induced by Injectable Solutions of CUR and RES

To show the cytotoxic effect induced by CUR, RES and the 75:25 combination of CUR:RES, A549 and BEAS cells (700,000 cells/well) were seeded in 6-well plates and treated with the injectable solutions for 48 h; a negative control (without polyphenols) and positive control with cisplatin were included. After incubation, cells were washed with saline solution and fixed with 4% paraformaldehyde for 20 min, followed by three washes with saline solution (five minutes each). Images were then acquired with a Nikon Eclipse TS2 microscope with a 20× Ph1 ADL objective using an AxioCam ERc 5S camera (ZEISS, Oberkochen, Germany).

### 2.8. Statistical Analysis

The solubility of CUR and RES (mg/mL) in different cosolvents, the hemolytic potential (%), the concentrations of CUR and RES (%) in the stability studies, and the cell viability (%) of the injectable formulations were subjected to an analysis of variance (ANOVA) followed by Tukey’s post hoc test to evaluate the differences between treatments (*p* < 0.05) using GraphPad Prism version 10.4.1 (GraphPad Software, San Diego, CA, USA). Data are expressed as mean ± standard error (SE). Before the ANOVA, compliance with normality (Shapiro–Wilk test) and homogeneity of variances (Leven’s test) were assessed. The 2^k^ and 3^k^ factorial designs were formulated and evaluated using Statgraphics Centurion XV, version 15.1.02 (StatisticalGraphics Co., Rockville, MD, USA). Effects were considered significant when *p* < 0.05.

## 3. Results

### 3.1. Solubility Studies of CUR and RES

The solubility of CUR and RES raw materials in different cosolvents used in injectable formulations is shown in Table 1. In the case of CUR, its solubility was better in both PEG 400 (3.62 mg/mL) and ethanol (3.60 mg/mL); in fact, no significant differences were found. However, its solubility in PG (2.12 mg/mL) was significantly lower than in PEG 400 and ethanol. In contrast, the solubility of RES was higher in PG (8.92 mg/mL) than in PEG 400 (2.39 mg/mL) and ethanol (2.05 mg/mL), being significantly lower in the latter.

### 3.2. Development and Selection of Injectable CUR Formulation

Figure 1 shows the effect of the factors (%) derived from the model coefficients for each response variable analyzed: CUR content (mg/mL) and pH. Factors such as CUR, PEG 400 and CUR–PEG 400 showed significant positive effects (*p* < 0.05) on the CUR content (Figure 1A); that is, the higher the proportion of PEG 400, the higher the CUR content in the formulations (mg/mL), and consequently, the proportion of CUR also tends to increase. The equation that describes the model was the following: CUR content (mg/mL) = 1.27748 + 0.28675(CUR) + 1.23788(PEG 400) + 0.2777(CUR × PEG 400). This model presented an adjusted coefficient of determination (adjusted R^2^) of 98.89%, which demonstrated that the variation in the model could be adequately explained and, therefore, had a high predictive capacity.

In the case of the pH response variable, only ethanol and PEG 400 showed significant positive effects (*p* < 0.05) on it, that is (Figure 1B), at high levels of these factors, an increase in the pH of the formulations was favored. Although CUR did not show a significant effect, it tends to slightly increase the pH. The equation that describes the model is pH = 6.49021 + 0.0677083(CUR) + 0.502708(PEG 400) + 0.167708(Ethanol), which adequately explained the variability in pH since the adjusted R^2^ was 93.88%.

Once the experimental design was analyzed, the optimal injectable formulation based on the highest amount of dissolved CUR consisted of 0.3% CUR, 30% ethanol, 60% PEG 400 and 9.7% water. However, as the pH of this injectable formulation was above 7.0, to increase the stability of CUR in solution, the percentage of water was replaced by a lactic acid–sodium lactate buffer solution, thus reaching a pH of 5.6. This prototype formulation was used for the following studies.

For further details on the experimental matrix of the 2^3^ factorial design, see Appendix A.

### 3.3. Development and Selection of Injectable RES Formulation

Figure 2 shows the effects (%) of the factors that explain the greatest variability in the model for each of the response variables—RES content (mg/mL) and pH—considering their coefficients. In the case of RES content (mg/mL) (Figure 2A), factors such as PG, PEG 400 and PG–PEG 400–ethanol presented significant positive effects on the response variable; that is, they tended to increase the RES content (mg/mL) in the injectable formulation. In contrast, the interactions of the factors PEG 400–ethanol and PG–PEG 400 tended to significantly decrease the RES content (negative effect). The equation describing the model involved in the RES content (mg/mL) in the formulations was as follows: RES content (mg/mL) = 89.5815 + 7.53333(PG) + 8.43889(PEG 400) − 12.4083(PG × PEG 400) − 8.70833(PEG 400 × Ethanol) + 11.775(PG × PEG 400 × Ethanol). With this model, 50.64% of the variability in the response could be explained.

As far as pH is concerned (Figure 2B), the interaction of the factors PEG 400–PEG 400–ethanol presented a significantly positive effect (increased the pH), while the interactions of PG–PEG 400–PEG–400 and PG–PG–ethanol presented negative effects; that is, they tended to reduce the pH. The model for this response variable explained 45.26% of the variability, and the equation was as follows: pH = 6.50741 + 0.255556(PG) − 0.15(PG × PEG 400) + 0.125(PEG 400 × Ethanol) + 0.0(PG2 × Ethanol) − 0.383333(PG × PEG 4002) − 0.03(PG × PEG 400 × Ethanol) + 0.241667(PG × Ethanol2) + 0.0(PEG 4002 × Ethanol).

After evaluating the experimental design, the optimal percentages of the injectable formulation were selected as 0.3% RES, 60.0% PEG 400, 20.0% ethanol and 0.8% NaCl, and the remainder comprised water (18.9%). The selection of the components of this injectable formulation was based primarily on the highest amount of dissolved RES. This prototype formulation was used for the following studies and had a pH of 6.7.

For further details on the experimental matrix of the 3^3^ factorial design, see Appendix A. 

### 3.4. Evaluation of the Hemolytic Potential of Injectable Formulations

The percentage of hemolytic potential of the injectable CUR and RES solution prototypes is shown in Table 2. The results indicated that the injectable RES solution had a significantly lower hemolytic potential compared to the injectable CUR solution, but both injectable solutions are considered non-hemolytic compared to the positive control (Extran^®^ MA 02), in which the erythrocytes underwent complete hemolysis (100.00%), since they presented hemolysis values < 10%. In the case of the negative control (Sörensen buffer), its hemolytic potential was null (0.00%).

### 3.5. Stability of Formulations Under Storage Conditions

The CUR and RES content in the individual injectable formulations and their combination (CUR (100.00%), RES (100.00%), CUR:RES (25:75), CUR:RES (50:50) and CUR:RES (75:25)) during the stability study is shown in Figure 3. In the injectable solutions of CUR (100.00%) and RES (100.00%), as well as in combinations of CUR:RES (50:50) and CUR:RES (75:25), the content of CUR and RES remained between 98.0% and 102.0%, with coefficients of variation of less than 2%; that is, there were no significant differences with respect to the storage time (0, 1 and 3 months). In contrast, in the combination of CUR:RES 25:75, the concentrations of both CUR and RES decreased significantly to 71.9% and 79.5%, respectively, only after 3 months of storage under refrigeration conditions, thus demonstrating that this combination of polyphenols compromises the stability of the injectable formulation. However, in the first month of storage, this formulation showed good stability, and the CUR and RES concentrations remained between 98.0% and 102.0%.

### 3.6. Cytotoxic Effects of Injectable Solutions of CUR, RES and Their Combination on Tumor and Non-Tumor Cells

The cytotoxic effects of injectable solutions of CUR, RES and their combination (25:75, 50:50 and 75:25) at four different doses on the adenocarcinoma A549 cell line are shown in Figure 4. Overall, injectable solutions of 100% CUR and RES had similar dose-dependent cytotoxic effects, but CUR decreased cell viability by between 0.3% and 23.5%, and RES between 1.2% and 34.7%. However, the combinations of CUR:RES 50:50 and 75:25 in the injectable solutions significantly reduced cell viability compared to the injectable CUR and RES (100%) solutions. These results suggest a synergistic effect between CUR and RES at concentrations of 0.020 mg/mL and 0.020 mg/mL (CUR:RES 50:50; CDI = 0.9) and 0.027 mg/mL and 0.010 mg/mL (CUR:RES 75:25; CDI = 0.7), respectively, since the CDI in both cases was less than 1. In this case, the combination of CUR:RES 75:25 reduced cell viability by 58.7%, while the 50:50 combination reduced it by 42.5%. Although the combination of CUR:RES 50:50 at concentrations of 0.010 mg/mL:0.010 mg/mL showed a significant reduction in cell viability compared to injectable solutions of 100% CUR and RES, the combination did not show a synergistic effect in reducing cell viability, but an additive one. In the other cases, the effect of the combination of CUR and RES was additive (CDI = 1). No combinations of injectable CUR and RES solutions showed an antagonistic effect.

In addition to the cytotoxicity evaluation of the formulations on A549 cells, cytotoxicity studies were performed on BEAS cells (Figure 5). This cell line was more susceptible to the effects of the injectable solutions than the A549 cells. In the 25:75 combination of the injectable solutions of CUR:RES, the concentrations of 0.009 mg/mL of CUR and 0.028 mg/mL of RES presented a synergistic effect in reducing cell viability, and at the following concentrations of CUR and RES (more diluted), the predominant effect was additive. In the case of the 50:50 combination (CUR:RES), only at the concentration of 0.005 mg/mL of CUR and 0.005 mg/mL of RES was an additive effect on cell viability observed, but at the other concentrations, the main effect was synergistic; that is, the cytotoxic effect on BEAS cells was enhanced. Finally, for the 75:25 combination (CUR:RES), specifically at the concentrations 0.007 mg/mL of CUR and 0.002 mg/mL of RES, a synergistic effect was observed since cell viability was reduced. Meanwhile, antagonistic effects prevailed at the other concentrations; that is, the combination of injectable solutions of CUR and RES tended to maintain or increase cell viability compared to CUR and RES (100%). However, while the percentage of CUR was higher in the injectable solution combinations, the cytotoxic capacity was also higher.

Since the combination of CUR:RES 75:25 presented the greatest cytotoxic capacity in A549 and BEAS cells, to support the results of the cytotoxicity study, cellular damage was evaluated by comparing the images of untreated cells (negative control) with those of cells treated either with cisplatin (positive control) or with CUR, RES or the CUR:RES combination (75:25) (Figure 6). In the case of cells exposed to cisplatin, CUR (0.027 mg/mL and 0.030 mg/mL in A549 and BEAS cells, respectively), RES (0.010 mg/mL in A549 and BEAS cells) and the combination of CUR:RES (75:25; 0.027 mg and 0.010 mg in A549 cells and 0.030 mg/mL and 0.010 mg/mL in BEAS cells), the cells were observed to be more elongated and compressed, with a considerable increase in cytoplasmic vacuoles, and in some cases, the cells had the appearance of blasts, with 30% cell detachment, compared to untreated cells (negative control).

## 4. Discussion

Cancer remains a major health concern, as it is the leading cause of death worldwide [25]. Among the different types of cancer, lung cancer has the highest incidence in men and the second highest in women [26,27]. Treatment for lung cancer initially includes surgery, followed by radiotherapy, chemotherapy or immunotherapy [28]. However, the treatment of choice depends on the stage of the disease; that is, in stages I and II, surgical resection of the tumor followed by adjuvant therapy is chosen, and in stages III and IV, treatment consists of chemotherapy or radiotherapy [29]. Standard or basic first-line chemotherapy for the treatment of non-small-cell lung cancer (NSCLC) is based on dual therapy with platinum-based drugs (cisplatin or carboplatin) and Etoposide and offers modest survival rates in patients with considerable disease progression [30,31]. However, this treatment can lead to adaptive responses of cells, which causes the development of resistance to therapy, that is, leads to tumor recurrence and disease progression [32]. In this regard, EGFR tyrosine kinase inhibitors (TKIs) such as Gefitinib, Erlotinib, Afatinib, Dacomitinib and Osimertinib are often the preferred treatment options [33]. Unfortunately, chemotherapy not only improves life expectancy but also produces side effects such as hair loss, mouth sores, loss of appetite and weight, nausea, vomiting, diarrhea and constipation. Therefore, it is necessary to make treatments more efficient and reduce their adverse effects [34,35].

In this context, naturally occurring polyphenolic compounds such as CUR and RES have received considerable attention due to their anticancer properties [36]. These polyphenols have been investigated as chemotherapy alternatives and adjuvants, either individually or in combination, to improve anticancer efficacy through synergistic effects, reducing doses and therefore adverse effects [37]. However, the consistency of their effects is hampered by multiple limitations, such as their poor aqueous solubility, reduced permeability, first-pass effect, low chemical stability and therefore reduced bioavailability [17,38]. To overcome these limitations, complexes of these phytobiotics with proteins, cyclodextrin inclusion complexes, emulsions, solid dispersions and, of course, nanoparticulate systems (polymeric micelles, nanogels, liposomes, nanoemulsions, polymeric nanocapsules, lipid nanoparticles) have been developed, the latter being the most studied at present [39,40]. Although nanotechnology has revolutionized the development of new drugs, one of its most important limitations lies in its scalability and reproducibility at an industrial level [41]. Therefore, in the present study, injectable solutions of CUR and RES were developed as a strategy to increase their solubility and stability, as well as to evaluate their cytotoxic potential individually and in combination on A549 and BEAS cells to establish possible synergistic effects and potential therapeutic alternatives to be evaluated in animal models of lung cancer.

For the development of the injectable solutions of CUR and RES, the solubility of CUR and RES in PEG 400, PG and ethanol was first evaluated since these are cosolvents authorized by the FDA [42] and have been shown to be safe alone or in combination when administered intravenously for 28 days at doses of 1 mL/kg in rats [43]. Only at doses higher than 2 mL/kg did small lesions appear at the injection site [43]. Furthermore, a study in dogs showed that the toxicity of PEG 400 was low at doses between 4.23 and 8.45 g/kg administered intravenously daily for 30 days [44]. In the case of propylene glycol, systemic toxicity has been reported to be very low at doses of 1 to 3 g/kg/day in rodents, whether administered orally or parenterally. However, at high doses (8–40 g/kg/day), central nervous system, hematological/hyperosmotic, cardiovascular and lactic acidosis effects can occur. In fact, like ethanol, propylene glycol has been shown to produce apoptotic neurodegeneration in the central nervous system of mice at doses of 2 g/kg [45]. Therefore, it can be said that the dosages and proportions of these cosolvents in injectable solutions are essential for safe formulations. As shown in Table 1, the solubility of CUR was highest in PEG 400, followed by ethanol and finally PG, but there were no significant differences between PEG 400 and ethanol. This was mainly due to the fact that the solubility of curcumin increased as the polarity of the medium was reduced; that is, as the dielectric constant of these cosolvents (PEG: 12.17; ethanol: 24.3; and PG: 32.0) decreases, solubility is favored [46]. Specifically, in the case of PEG 400, the polyhydric alcohols it contains in its chemical structure are capable of forming hydrogen bonds with the hydroxyl groups (–OH) and hydrogen atoms (–H) of curcumin, thus promoting its solubility. [46]. In contrast, RES presented higher solubility in PG (Table 1) because it has, as previously reported, much higher solubility in water and higher affinity for the aqueous phase compared to CUR [47]; that is, since PG had the highest dielectric constant (32.0), the solubility of RES was favored.

The injectable prototype solutions of CUR and RES were obtained and optimized by following a design of experiments and maintaining the percentages of the components within the recommended ranges: 11.2–67.0%, 10–60% and 15–50% for PEG 400, PG and ethanol, respectively [43,48,49]. In the case of the injectable solution of CUR, the pH of this injectable solution was reduced from 7.0 to 5.6 to increase the stability of CUR since it is well known that the solubility of CUR increases when the pH is greater, but its stability is compromised; that is, its degradation rate is high [50,51]. However, its safety is not compromised since, for injectable formulations for intravenous administration, pHs between 4.5 and 8.0 are permitted, with 7.4 being optimal [52]. Like CUR, resveratrol’s stability is exponentially increased at pHs greater than 6.8, but its solubility is not dependent on pH [16].

One of the factors that is crucial in parenteral formulations is hemolytic potential (rupture of red blood cells) since it can cause intravenous adverse effects, such as pain during administration, and release of hemoglobin can lead to vascular irritation, phlebitis, anemia, jaundice, kernalicterus, acute renal failure and, in some cases, death [22]. In order for a parenteral administration formulation to be considered non-hemolytic, it must have hemolysis values below 10% [22]; in the present study, the injectable CUR and RES solutions presented hemolysis values of 4.43 ± 0.05 and 2.45 ± 0.06, respectively, so they were considered non-hemolytic, and their administration can be considered safe. Furthermore, these injectable solutions of CUR and RES proved to be stable under refrigeration conditions (4 °C) for 3 months, which can be attributed to the components of the formulations since they prevent the hydrolytic degradation of both polyphenols [53]. However, when the combinations of injectable CUR:RES solutions (25:75, 50:50 and 75:25) were prepared and subjected to stability testing, only in the CUR:RES combination 25:75 did the amount of CUR and RES decrease significantly (to 71.9% and 79.5%, respectively). To date, there are no studies evaluating the stability of conventional or non-conventional formulations of CUR, RES or their combination for extended periods; several studies only focus on assessing the synergistic cytotoxic effects on different cell lines. Nonetheless, this behavior could be due to factors such as the reaction rate of these two polyphenols, their polarity and the concentration at which they are present in the formulation [54]. In fact, this last factor is one of the most important, as the rapid degradation of CUR [16,51,55] could affect the stability of RES, since it is present at 25% in the injectable solution. Therefore, it is suggested to avoid the 25:75 combination of CUR:RES in order not to compromise the stability of the formulation until the mechanism is fully elucidated.

The present study focused on evaluating the cytotoxic effect of injectable solutions of CUR, RES and their combination on tumor and non-tumor lung cells, with the aim of being able to use them as alternatives to the chemotherapeutics currently on the market or as adjuvants, considering that the FDA, the European Medicines Agency (EMA) and the World Health Organization (WHO) require that combination therapies be supported by preclinical studies that adequately justify their advantage over monotherapies [56]. The results showed that in the 50:50 (0.020 mg/mL:0.020 mg/mL) and 75:25 (0.027 mg/mL:0.010 mg/mL) combinations of CUR:RES, at high concentrations (Figure 4), the cytotoxic capacity was enhanced; that is, they presented a synergistic effect in reducing the viability of A549 cells (lung cancer tumor cells). These results are supported by studies that have shown synergistic effects of CUR and RES in different dosage forms in reducing the viability of colorectal cancer cells, cervical cancer cells, liver cancer cells, breast cancer cells and lung cancer cells, but only in the 50:50 combination [17,37,57,58]. However, the cytotoxic effect of CUR and RSV individually or in combination in injectable solutions on lung tumor or non-tumor cells has not been reported in the literature; evaluations have been performed by dissolving CUR and RES only in dimethyl sulfoxide [6,10]. Furthermore, regarding the combination of CUR:RES 75:25, there are no reports of its synergistic effect on cell lines; there is only one study in which a combination of 80:20 (similar to ours) produced strong synergistic activity against H_2_O_2_-induced oxidative stress in EAhy926 cells (human endothelial cells) [59]. It has been described that polyphenols can interact synergistically through a regenerative mechanism, meaning that one antioxidant regenerates the other [60]. However, for a synergistic effect to occur when combining polyphenols, the weaker one must regenerate the more potent one; otherwise, the effect is antagonistic [54]. In fact, RES is known to be capable of regenerating oxidized CUR to its reduced form, which has allowed its antioxidant activity to be enhanced [60]. The mechanism by which both polyphenols reduce cell viability is related to their ability to interfere with cell death, mainly apoptosis and autophagy; the modulation of transcription factors, protein kinases and cell cycle regulatory proteins; and the inhibition of angiogenesis [4,61]. Clearly, the doses and proportions of polyphenols at which the cytotoxic effects of injectable solutions were evaluated are important for establishing their effects and thus correctly selecting the formulation with the best performance [56,62].

Compared to A549 cells, BEAS cells (non-tumor lung cells) have been reported to be more susceptible to treatments [63]. In the present study, BEAS cells were around 50% to 70% more susceptible to treatments with polyphenols, mainly because A549 tumor cells are likely to have successive mutations in the proto-oncogenes involved in proliferation and/or cell death, making them more tolerant [64]. The results obtained demonstrated that the 50:50 combination of CUR:RES at different doses presented mainly synergistic effects, even at low doses (Figure 5). In contrast, in the 75:25 combination of CUR:RES, the presence of RES increased cell viability in the combination because it limits cell apoptosis, which leads to antagonistic effects with CUR [9]. However, it is a fact that the responses of both cell lines are related to the cell–treatment interaction, the composition of the culture medium and the susceptibility of the cell line. In this case, the calculation of selectivity indices would not be able to clearly establish the selectivity of the treatments since the IC50 in non-tumor cells (BEAS) would be lower compared to that in tumor cells (A549), which would lead to selectivity indices of less than 1 and, consequently, indicate that the treatments are generally cytotoxic, that is, non-selective [65]. Therefore, it is necessary to conduct studies in animal models to understand the effects obtained in vitro.

## 5. Conclusions

In conclusion, the present study proposes the use of intravenously injectable solutions of CUR and RES and their different combinations as potential therapeutic alternatives that should be evaluated in animal models of lung cancer, specifically the 75:25 combination of CUR and RES, as it exhibited synergistic cytotoxic effects on A549 cells, presenting a CDI of 0.7, which was lower than that of the 50:50 combination (CDI: 0.9). Likewise, the CUR:RES 75:25 combination showed antagonistic effects on BEAS cells, in contrast to the 50:50 combination, which showed synergistic effects. This may be related to the greater specificity of the 75:25 combination toward tumor cells (A549). Furthermore, the development of injectable solutions addresses the problems of reproducibility and scaling that nanotechnological systems present, since the processes for obtaining them are simple and easy to scale up to an industrial level. Further studies in rodents are being conducted to support the results obtained in cell cultures, since it is necessary to understand the pharmacokinetic behavior of individual and combined injectable solutions in order to assess their potential use in therapies or as adjuvants.

## Figures and Tables

**Figure 1 antioxidants-14-00983-f001:**
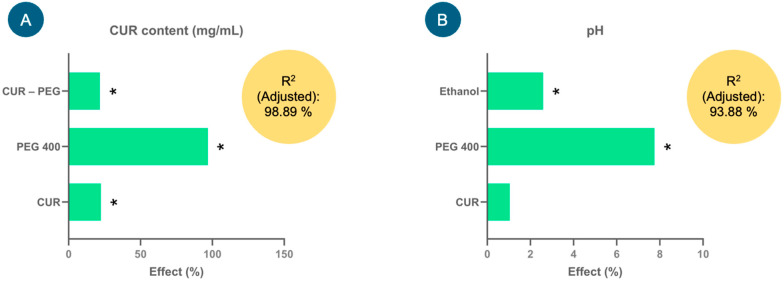
Effects of the factors (%) studied in the experimental design for the injectable curcumin (CUR) solution on the response variables, (**A**) CUR content (mg/mL) and (**B**) pH, used to select the injectable formulation. * Indicates significant effects, *p* < 0.05.

**Figure 2 antioxidants-14-00983-f002:**
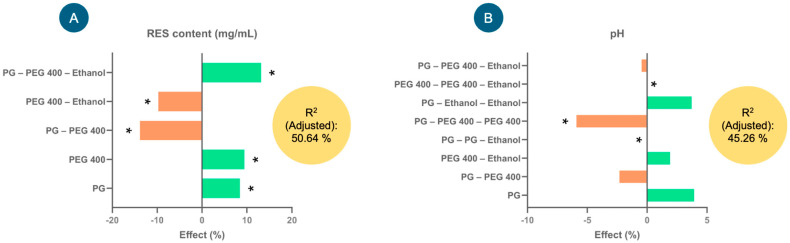
The effects of the factors studied in the experimental design for the injectable resveratrol (RES) solution on the response variables, (**A**) RES content (mg/mL) and (**B**) pH, used to select the injectable formulation. * Indicates significant effects (*p* < 0.05).

**Figure 3 antioxidants-14-00983-f003:**
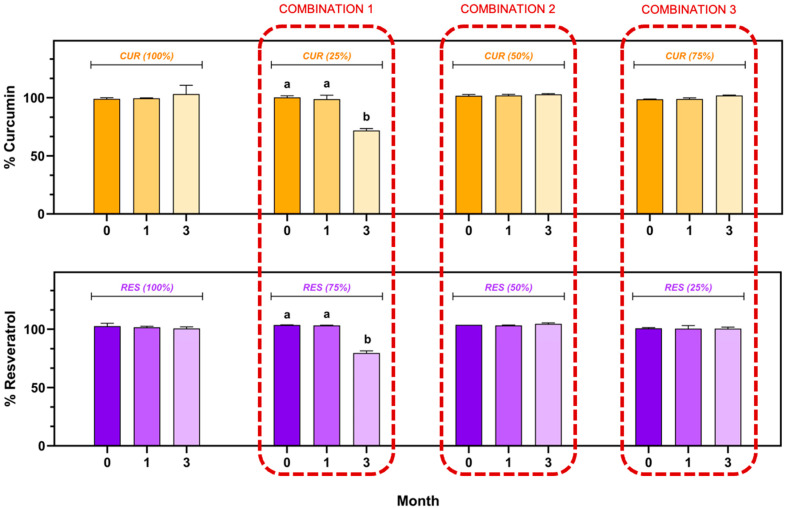
Stability of injectable solutions of curcumin (CUR), resveratrol (RES) and their combination under refrigerated storage conditions (5 ± 3 °C). Results are expressed as mean ± SE (n = 3). ^a,b^ Bars with different letters are considered significant (*p* < 0.05).

**Figure 4 antioxidants-14-00983-f004:**
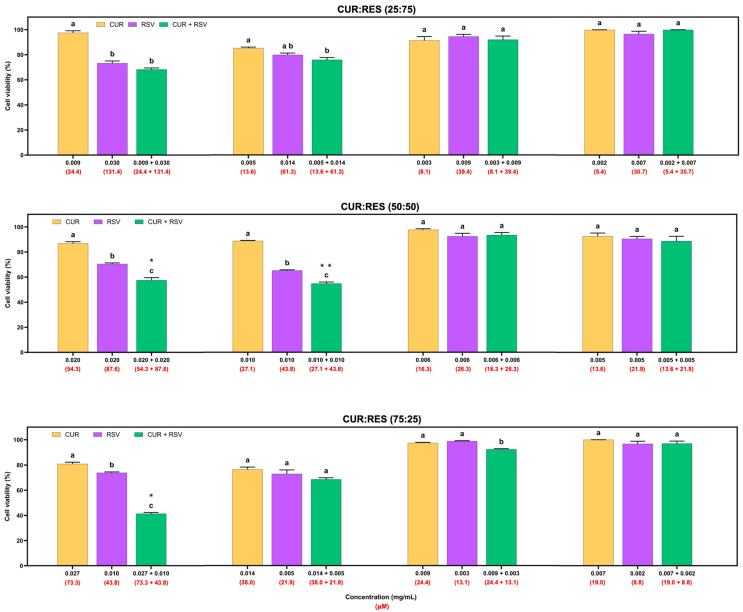
Cytotoxic effects of injectable solutions of curcumin (CUR), resveratrol (RES) and their combination (25:75, 50:50 and 75:25) on A549 cells after being exposed for 48 h. Results are expressed as mean ± SE of three independent experiments (n = 6). ^a–c^ Bars with different letters are considered significant (*p* < 0.05). * Indicates synergistic effect in CUR:RES combinations, and ** indicates additive effect of CUR:RES combination.

**Figure 5 antioxidants-14-00983-f005:**
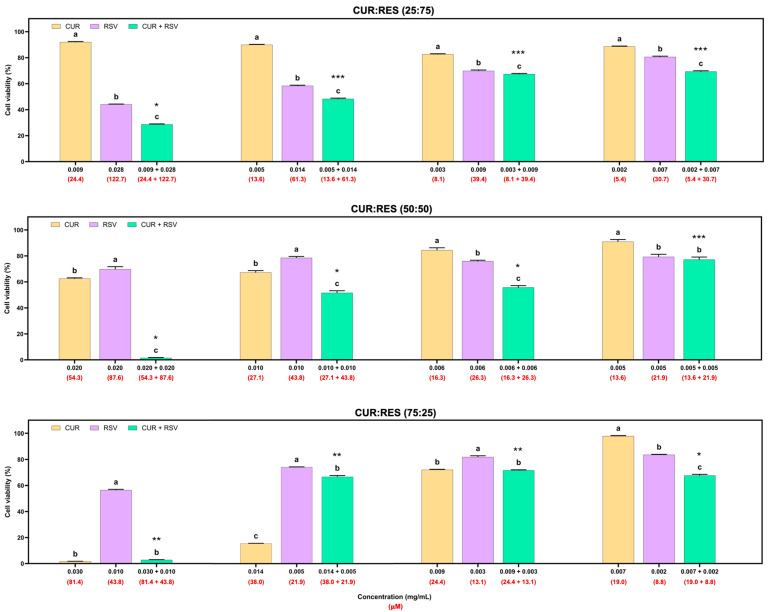
Cytotoxic effects of injectable solutions of curcumin (CUR), resveratrol (RES) and their combination (25:75, 50:50 and 75:25) on BEAS cells after being exposed for 48 h. Results are expressed as mean ± SE of three independent experiments (n = 6). ^a–c^ Bars with different letters are considered significant (*p* < 0.05). * Indicates synergistic effect of CUR:RES combination; ** indicates antagonistic effect of CUR:RES combination; and *** indicates additive effect of CUR:RES combination.

**Figure 6 antioxidants-14-00983-f006:**
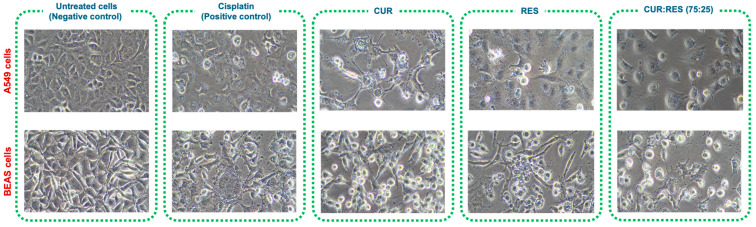
Representative images at 20× magnification of the cytotoxic effect of injectable solutions of curcumin (CUR), resveratrol (RES) and the combination of CUR:RES (75:25) on A549 and BEAS cells.

**Table 1 antioxidants-14-00983-t001:** Solubility of curcumin (CUR) and resveratrol (RES) raw materials in different cosolvents used in injectable formulations ^1^.

Cosolvent	Solubility of CUR (mg/mL)	Solubility of RES (mg/mL)
Propylene glycol (PG)	2.12 ± 0.06 ^b^	8.92 ± 0 06 ^a^
Polyethylene glycol 400 (PEG 400)	3.62 ± 0.04 ^a^	2.39 ± 0.04 ^b^
Ethanol	3.60 ± 0.06 ^a^	2.05 ± 0.03 ^c^

^1^ Results are expressed as mean ± SE, (n = 3). ^a–c^ Values within columns with different lowercase superscripts differ significantly (*p* < 0.05).

**Table 2 antioxidants-14-00983-t002:** Results of the hemolytic potential (%) of the prototype injectable CUR and RES solutions ^1^.

Injectable Formulation	Hemolytic Potential (%)	Interpretation
Sörensen buffer	0.00 ± 0.00 ^d^	Non-hemolytic
Extran^®^ MA 02	100.00 ± 0.00 ^a^	Hemolytic
CUR	4.43 ± 0.05 ^b^	Non-hemolytic
RES	2.45 ± 0.06 ^c^	Non-hemolytic

^1^ Results are expressed as mean ± SE (n = 3). ^a–d^ Values within columns with different lowercase superscripts differ significantly, *p* < 0.05.

## Data Availability

The databases used and analyzed during the current study are available from the corresponding authors.

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
