# Peer review of "Development and In Vitro Cytotoxicity Evaluation of Individual and Combined Injectable Solutions of Curcumin and Resveratrol Against Lung Cancer Cells"

_antioxidants, 2025, doi:10.3390/antiox14080983_

Round 1

Reviewer 1 Report

The manuscript presents a relevant and well-structured experimental study on the development and in vitro evaluation of injectable formulations of curcumin and resveratrol for potential cytotoxic applications. The topic is timely, and the methodology is generally sound. However, several important aspects need to be clarified or revised before the article can be considered for publication.

Dear authors,

We sincerely appreciate the opportunity to review your manuscript entitled “Development and In Vitro Evaluation of Injectable Solutions of Curcumin and Resveratrol as Potential Therapeutic Alternatives for Lung Cancer.” Your study addresses a highly relevant topic and presents a commendable experimental effort in developing parenteral solutions of polyphenols with potential cytotoxic activity. Nevertheless, after a thorough analysis, we believe that several substantial issues should be addressed before the manuscript can be considered for publication.

First and foremost, the manuscript presents a relevant in vitro exploration of the cytotoxicity of curcumin and resveratrol formulations against A549 and BEAS cells. However, the translational value of these findings is currently limited by the absence of a pharmacokinetic discussion on tissue distribution, metabolism, and absorption of these polyphenols via the proposed parenteral route. Since parenteral administration bypasses first-pass metabolism, it is essential to discuss biodistribution and potential metabolic fates of curcumin and resveratrol after systemic administration, particularly in the presence of excipients such as PEG 400 and ethanol.

Moreover, it is unclear whether the formulations are intended for intravenous, intramuscular, intradermal, or subcutaneous administration. The term “injectable” is broad and does not imply intravenous use by default. Each parenteral route carries distinct pharmacokinetic implications and safety profiles. Given the known systemic risks associated with intravenous administration of PEG 400 and ethanol—including local irritation, neurotoxicity, and potential organ toxicity—the omission of a clear definition of the administration route is a significant concern. We strongly recommend that the intended route be explicitly stated, and a toxicological rationale for the use of each excipient be included.

Regarding formulation aspects, we observed that the curcumin used in the study has a declared purity of 55.47%. However, the potential impact of the remaining 44.53%—which may include diverse degradation products, curcuminoids, or other impurities—was not discussed. We ask whether these components could have influenced the observed cytotoxicity or interfered with analytical quantification. In contrast, the standard reference for analytical calibration used curcumin of 65% purity, introducing a possible inconsistency. It would be important to clarify whether the experimental formulation was corrected for purity and whether the potential activity of non-curcumin components was considered.

The use of the Biopharmaceutical Classification System (BCS) to describe the solubility/permeability class of the compounds is only relevant for oral administration. Since the focus of the study is on parenteral delivery, the inclusion of BCS classification and the discussion of absorption enhancers like piperine (which is also used primarily for oral bioavailability improvement) lacks contextual relevance and should be reconsidered or redirected to emphasize solubility enhancement in parenteral settings.

From an experimental design perspective, we noticed that different factorial designs (2k and 3k) were employed for optimizing curcumin and resveratrol formulations. While the designs are valid individually, their use in parallel for comparative purposes raises concerns about standardization. If both active compounds were intended to be co-formulated, a unified experimental approach could have provided more consistent and integrated results. Alternatively, an explanation for using different designs should be provided to justify the approach.

Additionally, we would like to point out that the title and use of Boolean terminology such as “and” may imply that a combined formulation was developed throughout, whereas most of the study evaluates the compounds both separately and in combination. This nuance should be clarified to avoid misinterpretation.

Regarding the interpretation of results, we respectfully suggest a careful revision of the language used throughout the discussion, which currently appears overly adverbial and assertive without sufficient supporting data. Terms such as “alternative therapy” or “adjuvant potential” imply clinical readiness, yet no preclinical pharmacokinetics, in vivo efficacy, or toxicological data have been provided. Therefore, such extrapolations are speculative at this stage and must be softened or removed.

Although the formulation stability was addressed over a 3-month period under refrigerated conditions, it would be highly informative to include pH measurements during this period as well. Given that pH can significantly affect the stability of both curcumin and resveratrol, especially in aqueous solutions, such data are crucial to validate the robustness of the formulations. Furthermore, the decreased stability observed for the 25:75 CUR:RES combination lacks mechanistic explanation. We encourage the authors to discuss possible chemical interactions, oxidation reactions, or micellar disruptions that could account for this observed degradation.

With regard to cytotoxicity analysis, it is important to note that no calculation of selectivity index (SI) was provided to differentiate the effects between tumor and non-tumor cell lines. Since cytotoxicity against non-tumoral BEAS cells was also observed, this raises concerns about therapeutic selectivity. Including SI values would provide a more nuanced understanding of potential efficacy versus toxicity.

Furthermore, we encourage the authors to present concentrations in both mg/mL and mmol/L consistently, especially in Figures 4 and 5, to facilitate better comparison and reproducibility. The current data presentation could obscure concentration-dependent relationships across different experimental conditions.

From a mechanistic perspective, the manuscript would benefit from a deeper exploration of why synergistic effects between CUR and RES were observed. One potential route would be to incorporate in silico prediction tools such as PASS Online, Molinspiration Bioactivity Score, or the SEA Search Server. These tools can help hypothesize likely molecular targets, biological pathways, or transporter interactions (such as P-glycoprotein inhibition) which were not addressed in the current manuscript. In particular, modulation of efflux transporters like P-gp is a known mechanism through which polyphenols enhance cytotoxic drug uptake, and this aspect could enrich the discussion significantly.

Finally, we urge the authors to include a detailed discussion of cosolvent toxicity in the context of parenteral use. While PEG 400 and ethanol are commonly used, their intravenous administration is not universally safe, and dosage thresholds, infusion rates, and systemic exposure need to be discussed. If the intent is not intravenous but another parenteral route (e.g., intramuscular or subcutaneous), the manuscript should explain how local tolerability was considered.

In summary, the study presents relevant experimental results and a solid formulation effort; however, substantial revisions are required to enhance its scientific rigor and translational significance. We look forward to receiving a thoroughly revised manuscript that addresses the aforementioned points with due scientific depth and clarity.

Warm regards,

Author Response

Dear Reviewer 1

We thank you very much for the time you have spent on reviewing our manuscript. We have given full consideration to your comments and the manuscript that has been carefully revised and modified accordingly. Please refer to the point-by-point reply to your comments

Comment 1

The current title — “Development and In Vitro Evaluation of Injectable Solutions of Curcumin and Resveratrol as Potential Therapeutic Alternatives for Lung Cancer” — partially reflects the content of the article. However, some refinements are necessary to enhance its precision and avoid speculative or misleading interpretations. First, the phrase "potential therapeutic alternatives for lung cancer" may lead readers to assume that the study demonstrates a clinically relevant or preclinically validated therapeutic effect. Given that the manuscript presents only in vitro cytotoxicity data, without pharmacokinetic evaluation, in vivo efficacy, or toxicity profiling, the suggestion of a therapeutic alternative remains premature and speculative. A more cautious and accurate phrasing — such as “cytotoxic evaluation” or “preliminary in vitro assessment” — would better represent the scope and limitations of the study. Second, the title implies the development of a combined formulation of curcumin and resveratrol throughout. However, much of the study evaluates each compound separately, and only part of the experimental design focuses on their combinations. To reflect this more accurately, the title could indicate the evaluation of both individual and combined formulations, for example: “Development and In Vitro Cytotoxic Evaluation of Individual and Combined Injectable Solutions of Curcumin and Resveratrol against Lung Cancer Cells.” Such a revision would reduce interpretative ambiguity and align the title more closely with the study's actual design and findings.

Response 1

Thank you very much for the suggestion, and we believe the proposed title is the most appropriate. Therefore, it has been changed to the proposed one.

Comment 2

We sincerely appreciate the opportunity to review your manuscript entitled “Development and In Vitro Evaluation of Injectable Solutions of Curcumin and Resveratrol as Potential Therapeutic Alternatives for Lung Cancer.” Your study addresses a highly relevant topic and presents a commendable experimental effort in developing parenteral solutions of polyphenols with potential cytotoxic activity. Nevertheless, after a thorough analysis, we believe that several substantial issues should be addressed before the manuscript can be considered for publication.

First and foremost, the manuscript presents a relevant in vitro exploration of the cytotoxicity of curcumin and resveratrol formulations against A549 and BEAS cells. However, the translational value of these findings is currently limited by the absence of a pharmacokinetic discussion on tissue distribution, metabolism, and absorption of these polyphenols via the proposed parenteral route. Since parenteral administration bypasses first-pass metabolism, it is essential to discuss biodistribution and potential metabolic fates of curcumin and resveratrol after systemic administration, particularly in the presence of excipients such as PEG 400 and ethanol.

Response 2

We understand your concerns, but this study allowed us to select the best of the prototype formulations to conduct animal studies. However, we include in the conclusion that part of the limitations of our studies.

Comment 3

Moreover, it is unclear whether the formulations are intended for intravenous, intramuscular, intradermal, or subcutaneous administration. The term “injectable” is broad and does not imply intravenous use by default. Each parenteral route carries distinct pharmacokinetic implications and safety profiles. Given the known systemic risks associated with intravenous administration of PEG 400 and ethanol—including local irritation, neurotoxicity, and potential organ toxicity—the omission of a clear definition of the administration route is a significant concern. We strongly recommend that the intended route be explicitly stated, and a toxicological rationale for the use of each excipient be included.

Response 2

We agree with what you've said, but since this is a prototype formulation, we had in mind a formulation for intravenous or intramuscular administration, considering that the cosolvents selected for the formulations are acceptable for both routes of administration, as they are considered safe up to a certain percentage in the formulation and at certain doses. However, some toxicological implications caused by their use were placed in the discussion.

Comment 3

Regarding formulation aspects, we observed that the curcumin used in the study has a declared purity of 55.47%. However, the potential impact of the remaining 44.53%—which may include diverse degradation products, curcuminoids, or other impurities—was not discussed. We ask whether these components could have influenced the observed cytotoxicity or interfered with analytical quantification. In contrast, the standard reference for analytical calibration used curcumin of 65% purity, introducing a possible inconsistency. It would be important to clarify whether the experimental formulation was corrected for purity and whether the potential activity of non-curcumin components was considered.

Response 3

The impact of the remaining 44.53% that was not curcumin within the product was considered, so the amount of curcumin was adjusted considering its purity and it could be partially assumed that the effect is mainly due to curcumin. In fact, curcumin has been reported to be the curcuminoid with the best antioxidant properties compared to demethoxycurcumin and bisdemethoxycurcumin [1].However, we are aware that for further studies we need raw materials of higher purity.

The reason why a higher purity product was used for the analytical method was because a standard with purity greater than 99% was used and the 65% standard was used as a secondary standard for routine analysis. Furthermore, since the raw material for curcumin is somewhat expensive at Sigma Aldich, a local producer offers curcumin with a purity of 55.47% at a lower cost, which is why it was used for the development of the formulations.

[1] Jayaprakasha, G. K., Rao, L. J., & Sakariah, K. K. (2006). Antioxidant activities of curcumin, demethoxycurcumin and bisdemethoxycurcumin. Food chemistry, 98(4), 720-724.

Comment 4

The use of the Biopharmaceutical Classification System (BCS) to describe the solubility/permeability class of the compounds is only relevant for oral administration. Since the focus of the study is on parenteral delivery, the inclusion of BCS classification and the discussion of absorption enhancers like piperine (which is also used primarily for oral bioavailability improvement) lacks contextual relevance and should be reconsidered or redirected to emphasize solubility enhancement in parenteral settings.

Response 4

Thank you very much for the suggestion. In this regard, the use of curcumin absorption enhancers such as piperine has been deleted.

Comment 5

From an experimental design perspective, we noticed that different factorial designs (2k and 3k) were employed for optimizing curcumin and resveratrol formulations. While the designs are valid individually, their use in parallel for comparative purposes raises concerns about standardization. If both active compounds were intended to be co-formulated, a unified experimental approach could have provided more consistent and integrated results. Alternatively, an explanation for using different designs should be provided to justify the approach.

Response 5

As you mentioned, the designs to optimize the formulations were different for curcumin and resveratrol. This was because the optimization of the solutions was part of two projects carried out by undergraduate students. At the time, these two (somewhat different) designs were proposed. However, the optimized formulations exhibited good physicochemical characteristics and stability, so the second stage of this work was to evaluate their individual effects, but we also included the different combinations, as they exhibited good characteristics.

Each of the students mentored by different members of our research group considered solubility as the main factor. For this reason, in one of the designs, the concentration of curcumin was included at two levels and in the case of resveratrol, the concentration factor was kept constant, since the solubility of resveratrol is higher compared to that of curcumin.

Comment 6

Additionally, we would like to point out that the title and use of Boolean terminology such as “and” may imply that a combined formulation was developed throughout, whereas most of the study evaluates the compounds both separately and in combination. This nuance should be clarified to avoid misinterpretation.

Response 6

To avoid confusion about the fact that the formulations were developed with both polyphenols, it was clarified throughout the manuscript. Thank you very much for the suggestion.

Comment 7

Regarding the interpretation of results, we respectfully suggest a careful revision of the language used throughout the discussion, which currently appears overly adverbial and assertive without sufficient supporting data. Terms such as “alternative therapy” or “adjuvant potential” imply clinical readiness, yet no preclinical pharmacokinetics, in vivo efficacy, or toxicological data have been provided. Therefore, such extrapolations are speculative at this stage and must be softened or removed.

Response 7

Thank you very much for the suggestions. We believe that your comment is accurate given the results shown in the manuscript.

Comment 8

Although the formulation stability was addressed over a 3-month period under refrigerated conditions, it would be highly informative to include pH measurements during this period as well. Given that pH can significantly affect the stability of both curcumin and resveratrol, especially in aqueous solutions, such data are crucial to validate the robustness of the formulations. Furthermore, the decreased stability observed for the 25:75 CUR:RES combination lacks mechanistic explanation. We encourage the authors to discuss possible chemical interactions, oxidation reactions, or micellar disruptions that could account for this observed degradation.

Response 8

We understand the concern, and in the studies we are about to begin under accelerated conditions, we will consider pH measurement. At this time, it is impossible to include them as they were not considered and could be a limitation of the study. However, the mechanism by which stability could have been affected in the 25:75 combination was further discussed.

Comment 9

With regard to cytotoxicity analysis, it is important to note that no calculation of selectivity index (SI) was provided to differentiate the effects between tumor and non-tumor cell lines. Since cytotoxicity against non-tumoral BEAS cells was also observed, this raises concerns about therapeutic selectivity. Including SI values would provide a more nuanced understanding of potential efficacy versus toxicity.

Response 9

Thank you very much for your comment. Our initial objective was to determine whether the injectable formulations of curcumin and resveratrol, as well as their combinations, would exhibit cytotoxic effects on the two cell lines, but we did not establish IC50s. However, it could be said that the formulations are not selective because, as mentioned in the discussion, non-tumor cells (BEAS) were more susceptible to the treatment, meaning the IC50s would be lower compared to A549 (tumor cells). In this sense, the selectivity index would be less than 1, which is related to general (non-selective) cytotoxicity. We include what was commented in the discussion.

Comment 10

Furthermore, we encourage the authors to present concentrations in both mg/mL and mmol/L consistently, especially in Figures 4 and 5, to facilitate better comparison and reproducibility. The current data presentation could obscure concentration-dependent relationships across different experimental conditions.ç

Response 10

Thank you very much for your comment. Figures 4 and 5 have been modified and also show the concentrations in uM.

Comment 11

From a mechanistic perspective, the manuscript would benefit from a deeper exploration of why synergistic effects between CUR and RES were observed. One potential route would be to incorporate in silico prediction tools such as PASS Online, Molinspiration Bioactivity Score, or the SEA Search Server. These tools can help hypothesize likely molecular targets, biological pathways, or transporter interactions (such as P-glycoprotein inhibition) which were not addressed in the current manuscript. In particular, modulation of efflux transporters like P-gp is a known mechanism through which polyphenols enhance cytotoxic drug uptake, and this aspect could enrich the discussion significantly.

Response 11

We fully understand the comment, and as you rightly point out, it would be beneficial to perform the prediction in silico. Unfortunately, our research group lacks experience with this type of modeling, and third-party analysis would take between three and four weeks. In this regard, we try to reinforce the discussion with already published studies, hoping that they fulfill their perspective. Our apologies, and thank you for the suggestion.

Comment 11

Finally, we urge the authors to include a detailed discussion of cosolvent toxicity in the context of parenteral use. While PEG 400 and ethanol are commonly used, their intravenous administration is not universally safe, and dosage thresholds, infusion rates, and systemic exposure need to be discussed. If the intent is not intravenous but another parenteral route (e.g., intramuscular or subcutaneous), the manuscript should explain how local tolerability was considered.

Response 11

Thank you very much for your comment. The manuscript discusses the use of intravenous injectable solutions and the safety implications of the cosolvents used in their use in more detail.

Reviewer 2 Report

Natural products hold significant promise in pharmaceuticals; however, certain characteristics of these substances limit their development. This manuscript investigated the impact of formulation on the solubility and stability of curcumin (CUR) and resveratrol (RES) injection solutions, evaluated their cytotoxicity on A549 and BEAS cells. The study reveals a synergistic effect with the CUR:RES 75:25 combination, leading to reduced cell viability. This research offers insights into the application of CUR and RES injection solutions in lung cancer.

  1. Enhance the description of statistical methods, including multiple comparison methods for one-way ANOVA and statistical software used.
  2. Correct line 414 to read 0.005 mg/mL instead of 0.002 mg/mL.
  3. In Figure 4, the concentrations of CUR and RES at 0.010 mg/mL (CUR:RES, 50:50) may also warrant asterisk annotation.
  4. Clarify the criteria used to differentiate between synergistic, antagonistic, and additive effects based on the data.
  5. Why are the concentrations of CUR and RES not completely consistent in Figures 4 and 5?
  6. Figure 6 needs to be labeled with magnification factor, and consider adding non-representative images to supplementary materials.
  7. Justify the selection of CUR and RES, 75:25 over 50:50 in the conclusion.
  8. Expand the discussion section to include more in-depth comparisons and discussions with other relevant studies, as the current version primarily restates the results.
  9. If the title specifies "Lung Cancer," consider incorporating additional detection indicators in the manuscript.
  10. Line 36, determine whether a dose-dependent effect can be concluded based on Figures 4 and 5.
  11. Provide more detailed descriptions of methods, data, and results about sections 3.2 and 3.3.

Author Response

Dear Reviewer 2

We thank you very much for the time you have spent on reviewing our manuscript. We have given full consideration to your comments and the manuscript that has been carefully revised and modified accordingly. Please refer to the point-by-point reply to your comments

Comment 1

Enhance the description of statistical methods, including multiple comparison methods for one-way ANOVA and statistical software used.

Response 1

Thank you very much for the suggestion, the statistical analysis section was included in L.259-269.

Comment 2

Correct line 414 to read 0.005 mg/mL instead of 0.002 mg/mL.

Response 2

Thank you very much for the observation. In L.426, you can see the change.

Comment 3

In Figure 4, the concentrations of CUR and RES at 0.010 mg/mL (CUR:RES, 50:50) may also warrant asterisk annotation.

Response 3

We understand the suggestion made, but if we put asterisks in the additive effects, all combinations tested would have asterisks, except for the marked ones, which showed synergistic effects. In this sense, it was decided to only mention the two combinations with synergistic effects. However, if you think it is better as you indicate, we can do it.

Comment 4

Clarify the criteria used to differentiate between synergistic, antagonistic, and additive effects based on the data.

Response 4

As shown in the methodology, synergistic, antagonistic, or additive interactions between CUR and RES in injectable solutions were evaluated using the coefficient of drug interaction (CDI). CDIs less than 1 indicate synergistic effects, CDIs equal to 1 additive effects, and CDIs greater than 1 antagonistic effects. Formula:

CDI=C/(A×B)

where C represents the cell viability rate of the combination of CUR and RES in the injectable solutions; A and B represent the cell viability rates of CUR and RES, respectively.

In this sense, * means a CDI value less than 1 (synergistic), ** greater than 1 (antagonistic), and *** equal to 1 (additive).

Comment 5

Why are the concentrations of CUR and RES not completely consistent in Figures 4 and 5?

Response 5

We understand that the highest concentrations of CUR and RES in the different proportions were inconsistent, but this was due to the fact that when the samples were analyzed by UPLC, the actual concentrations varied slightly, but the change was not significant. For this reason, we retain the actual concentrations.

Comment 6

Figure 6 needs to be labeled with magnification factor, and consider adding non-representative images to supplementary materials.

Response 6

The magnification factor was set in Figure 6. Unfortunately, we do not have non-representative images as we only received the ones needed to include in this manuscript.

Comment 7

Justify the selection of CUR and RES, 75:25 over 50:50 in the conclusion.

Response 7

Some sentences have been added to better clarify the selection of the CUR:RES combination, 75:25. Thank you.

Comment 8

Expand the discussion section to include more in-depth comparisons and discussions with other relevant studies, as the current version primarily restates the results.

Response 8

The discussion was expanded as suggested. Thank you.

Comment 9

If the title specifies "Lung Cancer," consider incorporating additional detection indicators in the manuscript.

Response 9

The title of the manuscript was changed to:

"Development and In Vitro Cytotoxic Evaluation of Individual and Combined Injectable Solutions of Curcumin and Resveratrol against Lung Cancer Cells"

Comment 10

Line 36, determine whether a dose-dependent effect can be concluded based on Figures 4 and 5.

Response 10

Thanks for the feedback, the suggested changes were made.

Comment 11

Provide more detailed descriptions of methods, data, and results about sections 3.2 and 3.3.

Response 11

For further details on the experimental matrix of the 23 factorial design for CUR, see Supplementary Table 1.

For further details on the experimental matrix of the 33 factorial design for RES, see Supplementary Table 2.

Round 2

Reviewer 1 Report

I have reviewed the revised manuscript and confirm that the authors have adequately addressed all the comments and suggestions. In my opinion, the corrections are sufficient, and the manuscript is now suitable for publication.

None.

Reviewer 2 Report

The author has responded to the comments and revised the manuscript.

The author has responded to the comments and revised the manuscript.